# Automated MRI based pipeline for glioma segmentation and prediction of grade, IDH mutation and 1p19q co-deletion

**Milan Decuyper**                                          MILAN.DECUYPER@UGENT.BE
and **Stijn Bonte**                                         STIJN.BONTE@FLUIDDA.COM
and **Roel Van Holen**                                      ROEL.VANHOLEN@UGENT.BE
*Medical Image and Signal Processing (MEDISIP), Ghent University, Ghent, Belgium*

**Karel Deblaere**                                          KAREL.DEBLAERE@UGENT.BE
*Department of Radiology, Ghent University Hospital, Ghent, Belgium*

## Abstract

In the WHO glioma classification guidelines grade, IDH mutation and 1p19q co-deletion play a central role as they are important markers for prognosis and optimal therapy planning. Therefore, we propose a fully automatic, MRI based, 3D pipeline for glioma segmentation and classification. The designed segmentation network was a 3D U-Net achieving an average whole tumor dice score of 90%. After segmentation, the 3D tumor ROI is extracted and fed into the multi-task classification network. The network was trained and evaluated on a large heterogeneous dataset of 628 patients, collected from The Cancer Imaging Archive and BraTS 2019 databases. Additionally, the network was validated on an independent dataset of 110 patients retrospectively acquired at the Ghent University Hospital (GUH). Classification AUC scores are 0.93, 0.94 and 0.82 on the TCIA test data and 0.94, 0.86 and 0.87 on the GUH data for grade, IDH and 1p19q status respectively.

**Keywords:** Glioma, IDH mutation, 1p19q co-deletion, deep learning, MRI

## 1. Introduction

In the most recent WHO glioma classification guidelines three (genetic) markers, important for prognosis and optimal therapy planning, play a central role: WHO grade (glioblastoma, GBM versus lower-grade glioma, LGG), IDH mutation and 1p19q co-deletion (Louis et al., 2016; Yan et al., 2009; Weller et al., 2017). Biopsies to determine molecular information involve risks, are subject to sampling error and related to reduced OS compared to a wait-and-scan approach (Jackson et al., 2001; Wijnenga et al., 2017). Therefore, accurate non-invasive assessment of genetic mutations is desired. Most of the existing studies are not fully automatic, 2D, depend on expert opinion and are trained and evaluated on a small dataset (Yang et al., 2018; Choi et al., 2019; Akkus et al., 2017). In this study we propose a non-invasive fully automatic 3D pipeline to segment glioma and predict clinically relevant markers according to the most recent WHO guidelines. We collected a large dataset from multiple public databases and an independent dataset from our University Hospital (UH) to test generalization.

## 2. Materials and Methods

### 2.1. Data and Pre-processing

To acquire a large dataset, data was collected from multiple public databases: the TCGA-GBM (Scarpace et al., 2016), TCGA-LGG (Pedano et al., 2016) and 1p19qDeletion (Erickson et al., 2017) collections on The Cancer Imaging Archive (TCIA) (Clark et al., 2013) and the BraTS 2019 dataset (only patients not already included in the TCGA collections) (Menze et al., 2015; Bakas et al., 2017). Inclusion criteria were: a histologically proven glioma of WHO grade II, III or IV and the availability of preoperative T1ce MRI together with a T2 and/or FLAIR sequence of sufficient quality. In total 628 patients were included with known WHO grade. IDH mutation status was available for 380 patients and 1p19q co-deletion status for 280 LGG patients. Additionally, data was retrospectively acquired at our university hospital with permission of the local ethics committee (registration number withheld in anonymous version). We collected data from 110 patients with known WHO grade (61 GBM). IDH and 1p19q co-deletion status was determined for 86 (32 IDHmut) and 40 (12 co-deleted) patients respectively.

All MRI were co-registered, interpolated to 1 $mm^3$ voxel sizes, skull-stripped and independently normalized by subtracting the mean and dividing by the standard deviation. Tumor segmentation was done using a 3D U-Net (similar to Isensee et al. (2019)), trained on the BraTS 2019 training dataset and evaluated on the validation set by the online evaluation platform (https://ipp.cbica.upenn.edu). By randomly setting channels to zero during training (while making sure that at least the T1ce and a T2 or FLAIR sequence remained), the network shows increased robustness to missing modalities. This is beneficial as not all four MRI (T1, T1ce, T2, FLAIR) are available for every patient. An average whole tumor dice score of 90% is achieved based on all four MRI, 0.89 with only T1ce and FLAIR and 0.87 with T1ce and T2. This is close to performance of state-of-the art algorithms of the BraTS 2019 challenge according to the validation leaderboard (Bakas and Sako, 2019).

### 2.2. Multi-task Classification

After segmentation, a 3D tumor ROI (bounding box) is extracted and used as input to the subsequent classification network (Figure 1). The adaptive average pool layer allows the network to process different ROI input sizes hence no resizing to a fixed shape is required. The network is trained to simultaneously predict WHO grade, IDH mutation and 1p19q co-deletion status. This so-called multi-task learning helps the network to learn features that are relevant for multiple tasks and reduces the risk of overfitting. Moreover, as not all ground truth labels are available for every patient, multi-task learning allows us to train one network on all data instead of training separate networks for each task on a smaller dataset. The network is implemented in PyTorch and trained with AdamW optimization ($lr_{init} = 1 \cdot 10^{-5}$), L2 weight decay of $10^{-2}$, batch size of eight and focal binary cross-entropy loss on an Nvidia RTX 2080Ti. The loss is calculated for each task on all samples in the batch with known ground truth labels and averaged to a global loss. The 628 patients are split into a training set of 458 (264 GBM vs. 194 LGG, 123 IDHmut vs. 87 IDHwild and 83 1p19qDel vs. 100 1p19qIntact), a validation set of 70 (27 GBM, 41 IDH mut and 20 1p19qDel) and a test set of 100 (46 GBM, 48 IDHmut and 30 1p19qDel) patients.

The dataset was augmented with random flipping, axial rotations, intensity scaling, elastic transform and setting input channels to zero as was done to train the segmentation network. For patients in the validation and test set, all ground truth labels were available and test patients were not used in the training set of the segmentation network in order to evaluate the system on new cases that both the segmentation and classification stages have never seen before. Data from the GUH was used to evaluate the performance on an entirely independent dataset.

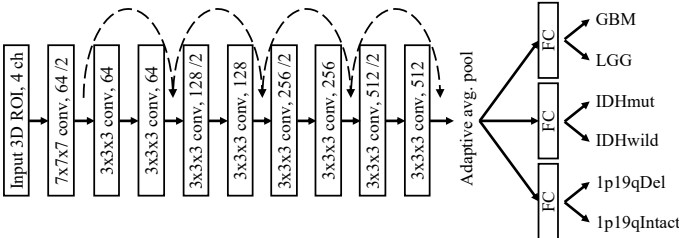

Figure 1: Schematic illustration of the classification architecture. Every convolutional layer is succeeded with instance normalisation and a ReLU activation

## 3. Results and Discussion

The AUC, accuracy, sensitivity and specificity scores on the TCIA and GUH test data are included in Table 1. The sensitivity scores indicate the percentage of GBM, IDHmut and 1p19qDel cases that are correctly classified as such. For binary grade prediction, very high accuracies of 90% are achieved on both the TCIA and GUH test data. This shows that the network is able to accurately distinguish GBM from LGG and generalizes well to unseen data from different institutions. The IDH prediction performance is high on the TCIA test set (AUC of 0.94) but lower on the GUH data (AUC of 0.86). Especially a lower specificity of 70% compared to 88% is observed. This difference might be because for the GUH data, IDH status was assesed through immunohistochemistry (IHC) while for the TCGA data gene sequencing was used. However, a negative IDH status using IHC does not necessarily mean an IDH wildtype tumor (Louis et al., 2016). Hence some IDH mutant astrocytoma might be missed with IHC resulting in more false positives of the model and thus a lower specificity. For 1p19q status we only included LGG cases as this marker is only considered for those patients in the WHO guidelines. Including the GBM cases (all 1p19qIntact) would increase the overall prediction accuracy but would introduce a large data imbalance and thereby decrease the performance for LGG cases. The GUH dataset only contains 12 1p19q co-deleted cases which might be too small to obtain reliable performance estimations. Depending on the classification threshold, the sensitivity for 1p19q status can also be optimized. For example, with a threshold of 0.45 the sensitivity on the GUH dataset increased to 75% with the same specificity.

Table 1: Classification results on both the TCIA and University Hospital test data

| TCIA \| GUH test data | AUC | Acc. | Sens. | Spec. |
|---|---|---|---|---|
| GBM vs. LGG | 93.3 \| 94.0 | 90.0 \| 90.0 | 93.5 \| 90.1 | 87.0 \| 89.8 |
| IDH status | 94.0 \| 86.2 | 89.0 \| 75.6 | 89.6 \| 84.4 | 88.5 \| 70.4 |
| 1p19q co-deletion | 82.1 \| 86.6 | 83.3 \| 75.0 | 86.7 \| 58.3 | 79.2 \| 82.1 |

## 4. Conclusion

In conclusion, we developed a fully automatic 3D pipeline to segment glioma and non-invasively predict important (molecular) markers according to the WHO classification guidelines with high diagnostic performance. The networks were trained on a large multi-institutional database and evaluated on an independent dataset which demonstrated the robustness and generalizability of the algorithm.

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
