# OpenReview forum: "Automated MRI based pipeline for glioma segmentation and prediction of grade, IDH mutation and 1p19q co-deletion"
_MIDL.io/2020/Conference — MIDL 2020_

### Official Review · AnonReviewer1 · 2020-03-05
**The authors proposed a deep learning based algorithm for brain tumor segmentation with prediction of grade, IDH mutation and 1p19q co-deletion.**

**Rating:** 3
**Confidence:** 5

**Review:**

The authors proposed a deep learning based algorithm for brain tumor segmentation with prediction of grade, IDH mutation and 1p19q co-deletion. The results are promising. However, there are two questions:
1. The authors mentioned that the network was trained and evaluated on a large heterogeneous dataset of 628 patients, collected from The Cancer Imaging Archive and BraTS 2019 databases. For my understanding the BraTS data may also include some data from The Cancer Imaging Archive. I am wondering if there will any data leaking during training and testing.
2. For the BraTS data, the author should refer to the latest benchmark: arXiv:1811.02629

---

### Official Review · AnonReviewer3 · 2020-03-11
**Nice evaluation of a multi-task learning approach for prediction of glioma grade and molecular characteristics**

**Rating:** 3
**Confidence:** 5

**Review:**

This short paper proposes a method for classification of glioma grade, IDH mutation, and 1p19q co-deletion, and trains/evaluates it on a fairly large dataset. The method also performs a segmentation, using a standard 3D U-Net trained on BraTS 2019 training data, with dropout on MRI sequences, to make the network robust to missing sequences in the application phase.
Strengths:
- A reasonable sized test set (100 patients).
- The pipeline seems engineered well.
- The classification accuracies are quite high.
Weaknesses:
- It is not clear how the tumor segmentation is exactly used in the classification network. Do you only use it to define a bounding box for the region of interest? Or do you mask the original image and set all pixels outside the segmentation to zero, for example?
- Section 3: "For 1p19q status we only included LGG cases" -> it's not clear whether you did this for the train or test set, or both.
- Confidence intervals should be given for the classification results in Table 1.
- Too many decimals are given in Table 1.
- Section 1: the relation to this work also could be discussed: https://www.ncbi.nlm.nih.gov/pubmed/31548344

---

### Official Review · AnonReviewer2 · 2020-03-13
**Automated MRI based pipeline for glioma segmentation and prediction of grade, IDH mutation and 1p19q co-deletion**

**Rating:** 2
**Confidence:** 5

**Review:**

This paper utilized the U-net to segmentation the glioma regions and then utilize the multi-task classification model to classify the corresponding grade, IDH mutation, and 1p19q co-deletion. The task of this paper is interesting. The technical novelties of this paper is low and this paper is an application-based model.

---

### Official Review · AnonReviewer4 · 2020-03-16
**Nice paper - interesting clinical problem, well explained experiments, good evaluation**

**Rating:** 4
**Confidence:** 4

**Review:**

quality: This is a well-written paper which tackles and interesting clinical problem, has a well-described framework and experiments and does a good evaluation.

clarity: Of course more details would be nice, but considering the brevity of the submission framework for short papers, the paper is very clear.

originality: I am not aware of the background clinical literature in this area, but it seems a novel application.

significance: The significance of the clinical solution is high and the presented algorithm seems to perform well enough to actually be a possible solution in the future.

pros:
- the paper has a very nice twist of making the network robust, but excluding certain image modalities during training
- the multi-task learning approach to learn the labels all at the same time is also very appropriate for this problem
cons:
- all abbreviations (OS, T1ce) should be introduced, not everyone has the same background in the clinic or in MRI to understand this
- why are there so different balances in training, validation and testing data? why were they not all divided in the same way in terms of cases?

CAVEAT: The authors state themselves in the abstract: "This short paper only contains a brief summary and selection of results from a manuscript that will shortly be submitted to Neuro-Oncology." Is this allowed according to MIDL guidelines?

---

### Meta-Review · Area_Chair1 · 2020-04-08
**MetaReview of Paper116 by AreaChair1**

**Rating:** 2

**Metareview:**

There are three reviews all in favour of the paper. One brief evaluation is rating it as 'weak reject'. There were critical questions raised by several reviewers, i.e., about relation to prior work and - a very important one - about leakage between training and testing data (R1).

The authors decided not to respond to it.  This is a critical matter as the high performances are the major innovation of the paper.

To this end, I would side with the critical reviewer and consider it as an application paper that can be accepted in case applications should be of particular interest at MIDL2020. Otherwise, I would rather consider it to be a weak reject.

**Paper Type:**

validation/application paper

---

### Decision · Program_Chairs · 2020-04-11

**Decision:**

Accept

**Comment:**

Taking all information into account, it was determined that the paper was accepted based on its merit.